# Phenotypic and Genotypic Analysis of Antimicrobial Resistance in *Escherichia coli* Recovered from Feedlot Beef Cattle in Australia

**DOI:** 10.3390/ani12172256

**Published:** 2022-08-31

**Authors:** Yohannes E. Messele, Mauida Alkhallawi, Tania Veltman, Darren J. Trott, Joe P. McMeniman, Stephen P. Kidd, Wai Y. Low, Kiro R. Petrovski

**Affiliations:** 1The Davies Livestock Research Centre, School of Animal and Veterinary Sciences, University of Adelaide, Adelaide, SA 5005, Australia; 2The Australian Centre for Antimicrobial Resistance Ecology, University of Adelaide, Adelaide, SA 5005, Australia; 3Meat & Livestock Australia, Level 1, 40 Mount Street, North Sydney, NSW 2060, Australia

**Keywords:** antimicrobial resistance genes, antimicrobial resistance surveillance, cattle feedlot, cattle slaughterhouse, multidrug resistance

## Abstract

**Simple Summary:**

Antimicrobial resistance (AMR) is a major concern for animal and human health. The use of antimicrobials is the main factor contributing to the development of AMR in food-producing animals but is unlikely to be the only factor. It is important to determine if antimicrobial use in a feedlot setting contributes to the overall resistance burden and what proportion of animals may already harbour resistant bacteria at feedlot entry. This project aimed to assess the level of AMR in *E. coli* isolated from beef cattle in South Australia at two time points, the entry and exit (at slaughter) from the beef feedlot. AMR frequency in *E. coli* isolated from entry compared to exit increased for tetracycline (0 to 17.8%), ampicillin (0.7 to 5.4%), streptomycin (0.7 to 4.7%), and sulfisoxazole (0 to 3.9%). Therefore, the regular inspection of these bacteria and their resistance determinants in food animals would be crucial to tracking changes in AMR and applying control mechanisms.

**Abstract:**

This study investigated the antimicrobial resistance (AMR) profile of fecal *Escherichia coli* isolates from beef cattle (n = 150) at entry and exit from an Australian feedlot. Sample plating on MacConkey agar and Brilliance ESBL agar differentiated generic from extended-spectrum β-lactamase (ESBL)-producing *E. coli*, respectively. Resistance profiles were determined by minimum inhibitory concentration (MIC) testing and further analyzed by whole-genome sequencing (WGS). At entry, the prevalence of antimicrobial resistance to amoxicillin/clavulanic acid, ampicillin, streptomycin, and trimethoprim/sulfamethoxazole was very low (0.7%, each). At the exit, the resistance prevalence was moderate to tetracycline (17.8%) and low to ampicillin (5.4%), streptomycin (4.7%), and sulfisoxazole (3.9%). The most common AMR genes observed in phenotypically resistant isolates were *tet(B)* (43.2%), *aph(3″)-Ib* and *aph(6)-Id (32.4%)*, *bla_TEM-1B_*, and *sul2* (24.3%, each), which are responsible for resistance to tetracyclines, aminoglycosides, β-lactams, and sulfonamides, respectively. The ESBL-producing *E. coli* were recovered from one sample (0.7%) obtained at entry and six samples (4.0%) at the exit. The ESBL-producing *E. coli* harbored *bla_TEM_* (29.7%), *bla_CTX m_
*(13.5%), and *bla_CMY_* (5.4%). The resistance phenotypes were highly correlated with resistance genotypes (*r* ≥ 0.85: *p* < 0.05). This study demonstrated that *E. coli* isolated from feedlot beef cattle can harbour AMR genes, but the low incidence of medically important resistance reflected the prudent antimicrobial use in the Australian industry.

## 1. Introduction

The use of antimicrobials for therapeutic and non-therapeutic purposes in animals plays an essential role in the development and selection of antimicrobial-resistant bacteria [1]. In addition, it is clear that other factors affect antimicrobial resistance (AMR) emergence and spread in animals, including environment, farm type, animal age category, management, and others yet to be confirmed [2]. The development of AMR through the acquisition of AMR genes (ARGs) assists pathogenic bacteria to overcome antimicrobial therapies and persist in competitive environments such as the gastrointestinal tract [3,4]. The development and spread of AMR, and the subsequent transfer of multidrug resistance among different bacteria via mobile genetic elements such as plasmids, is creating a considerable global problem of significance to both animal and human health [5]. Antimicrobial-resistant bacteria (or their ARGs) could potentially be transmitted from animal to human (or vice versa) through both direct contact and environmental contamination [6]. Whilst genetic similarities in *E. coli* plasmids isolated from animals and humans have been reported in some studies, the degree and significance of direct and indirect host-to-host AMR transfer are currently unclear in the published literature [7,8].

Carriage of antimicrobial-resistant *E. coli* by food-producing animals could be a potential risk factor for the acquisition of ARGs by medically important bacteria, especially if animal-derived commensals survive both pre-slaughter harvest intervention and cooking processes and then proliferate and/or transfer their ARGs to human pathogens under further selection pressure [9]. The risk of antimicrobial use in livestock contributing to human medical AMR needs to be set in the context of the probability of these factors occurring. Among AMR risks in food-producing animals, those posed by extended-spectrum β-lactamase (ESBL)-producing Enterobacteriales are of major significance, given that ESBLs are often associated with multidrug-resistant (MDR) infections and extended-spectrum β-lactams are often used as first-line therapies for treating Gram-negative sepsis in humans [10,11].

The increased use of third-generation cephalosporins such as ceftiofur may be linked with the emergence of ESBL-producing *E. coli* in food-producing animals [12,13], and given that most ESBLs are plasmid-derived, ESBL-positive *E. coli* frequently harbours genes encoding AMR to other classes of antimicrobials [14]. Hence, a comprehensive and careful investigation of AMR with a focus on ESBLs in both pathogens and commensals isolated from food-producing animals is necessary. Most previous studies on AMR in *E. coli* recovered from beef cattle have investigated resistance prevalence at a specific point in time, either on the farm or at the abattoir, but there are limited longitudinal studies that have sampled the same animal over longer time periods in the beef supply chain. For instance, research from Canada showed a significant increase in the level of antimicrobial-resistant *E. coli* recovered from the abattoir compared to the arrival into the feeding program [15]. Within Australia, AMR studies on *E. coli* isolated from food animals at slaughter have found a very low level of resistance generally and an absence of resistance to critically important antimicrobials used in human medicine [16,17].

Australia has antimicrobial management guidelines and the National Antimicrobial Monitoring System (AURA) to control and prevent antimicrobials [18]. Australia has followed a strict approach to the registration of antimicrobials for use in food-producing animals, for example, medically important antimicrobials in the fluoroquinolone class have never been permitted for use in food-producing animals [19]. However, macrolides, tetracyclines, and the third-generation cephalosporin ceftiofur are registered for use in cattle under veterinary prescription and are the main antimicrobials used to treat bovine respiratory disease (BRD) in beef feedlots in Australia [20]. To date, there have been no longitudinal studies conducted in Australian feedlot cattle that assess the risk of AMR acquisition during the supplementary feeding period (when BRD occurs more commonly, requiring antimicrobial treatment) that could potentially enter the food supply. Therefore, the objective of the present study was to assess the level of AMR in commensal *E. coli* recovered from cattle at entry into an Australian beef feedlot compared to the end of the feeding program during carcass processing at the abattoir.

## 2. Materials and Methods

### 2.1. Study Area and Animals

A longitudinal study was carried out to determine the prevalence and AMR profile of *E. coli* isolated from 150 randomly selected cattle at entry into a feedlot (located in South Australia with a total capacity of 17,000 head) and again from the same animals post-slaughter. The 150 cattle originated from three vendors (Location A: n = 82; Location B: n = 54; and Location C: n = 14). The cattle averaging 405 kg and the age of <2 years at feedlot entry were fed for a 90-day feeding period a diet consisting of variable proportions of barley, lucerne hay, oaten hay, lupins, almond hull, and concentrate supplements. The breeds of cattle were Angus, Hereford, Santa Gertrudis, Shorthorns, and their crosses. All cattle were housed in a single pen (the target pen). A tetracycline-based product was used for the metaphylaxis of the cattle arriving at the feedlot from high-risk sources (e.g., cattle bought from saleyards); however, it was not used on the target pen. If a clinically ill individual animal from the target pen was identified, it was transferred to the hospital pen for the duration of the treatment period, and then returned to the target pen when recovered. In this way, a total of 13 of the 150 cattle (8.7%) were treated therapeutically with antimicrobials during the study. Individual antimicrobial treatments included long-acting injections of either tulathromycin (n = 10), ceftiofur (n = 2) or oxytetracycline (n = 1). At the conclusion of the feeding period, the cattle were trucked approximately 150 km to an abattoir. The cattle remained in lairage for less than 4 h with access to clean water prior to slaughter.

### 2.2. Sample Collection

Using a single-use rectal glove, a fecal sample (approx. 15 g) was collected from the rectum of each incoming animal (n = 150) just before entry into the feedlot (i.e., at feedlot induction). After the 90-day feeding period, transport to the abattoir, slaughter, and evisceration, a second fecal sample was obtained from the same animal by inserting a sterile swab into an incision cut directly into the rectum. The swab was then immediately placed in Ames transport media and stored at 4 °C (Copan, Italy). The carcass swab samples were collected immediately after evisceration from both the flank and brisket (carcass site 1) and hip/round areas (carcass site 2) using sterile Puritan’s sampling swabs (Adelab Scientific, Australia). Each swab was charged by wiping an area of approximately 100 cm^2^ in a vertical and horizontal crossing pattern at each carcass site. All fecal samples and carcass swabs were then transported to the laboratory at 4 °C.

### 2.3. Bacterial Isolation

The isolation of *E. coli* was carried out following the method described by Kidsley, et al. [21]. Briefly, 10 g of feces from the feedlot entry samples or the post-slaughter rectal swabs themselves were directly added into 7 mL of sterile 0.1% buffered peptone water in a 50 mL falcon tube and vortexed for 30 s. A sterile cotton tip applicator was then used to apply an aliquot of the suspension onto MacConkey agar (MAC) for coliform selection and Brilliance ESBL agar (ESBL agar; Thermofisher Scientific, Victoria, Australia) to screen for ESBL-producing *E. coli*. The aliquots were streaked out for single colonies using a sterile loop and incubated at 37 ± 2 °C for 24 h. The carcass site swabs were placed in 5 mL peptone water diluent and homogenised for 30 s, and 100 µL aliquots were plated and streaked out [22] onto MacConkey and Brilliance ESBL agars (Thermofisher Scientific Australia) as described above. After incubation, single pink and blue colonies representative of the dominant morphology were picked from MAC and ESBL agar, respectively, and subcultured onto sheep blood agar incubated at 37 ± 2 °C for 24 h. All suspected *E. coli* isolates were definitively identified by matrix-assisted laser desorption ionisation time-of-flight (MALDI-TOF) mass spectrometry (Biotyper, Bruker Daltonik GMBH, Germany). All confirmed isolates were stored at −80 °C in tryptone soy broth plus 20% glycerol. Hereafter, the term generic *E. coli* (GE) will be used to indicate bacteria recovered from MAC and ESBL-producing *E. coli* (EE) for those recovered from ESBL agar.

### 2.4. Antimicrobial Susceptibility Testing

All *E. coli* isolates were subjected to antimicrobial susceptibility testing using commercially prepared broth microdilution panels to obtain minimum inhibitory concentration (MIC) values following Clinical and Laboratory Standards Institute (CLSI) [23] and European Committee on Antimicrobial Susceptibility Testing (EUCAST, 2020) standard guidelines. For this purpose, the standard Sensititre National Antimicrobial Resistance Monitoring System [24] (NARMS) Gram-negative CMV3AGNF MIC Plate was used. The following 14 antimicrobials were tested: amoxicillin/clavulanic acid, ampicillin, azithromycin, cefoxitin, ceftiofur, ceftriaxone, chloramphenicol, ciprofloxacin, gentamicin, nalidixic acid, streptomycin, sulfisoxazole, tetracycline, and trimethoprim/sulfamethoxazole (Trek Diagnostic Systems, Thermofisher Scientific, UK)*. E. coli* ATCC 25922, *E. coli* ATCC 35218, and *Pseudomonas aeruginosa* ATCC 27853 were included as the quality control strains. The inoculation and incubation were carried out following the manufacturer’s guidelines. The MIC test ranges and clinical breakpoint cut-off values for each antimicrobial are shown in Table 1. Based on these breakpoints, the isolates were classified as either susceptible or resistant. The resistance profile was categorized as MDR if the isolate exhibited resistance to one or more antimicrobials in three or more antimicrobial classes [25]. The AMR frequencies were described as rare: <0.1%; very low: 0.1% to 1.0%; low: >1% to 10.0%; moderate: >10.0% to 20.0%; high: >20.0% to 50.0%; very high: >50.0% to 70.0%; and extremely high: >70.0%; according to the European Food Safety Authority (EFSA) and the European Centre for Disease Prevention and Control (ECDC) [14].

### 2.5. DNA Extraction and Whole Genome Sequencing

All antimicrobial-resistant (n = 33, including seven extended-spectrum β-lactamase-producing isolates) and four susceptible isolates were selected for whole-genome sequence (WGS) analysis. Genomic DNA was extracted with a QIASymphony Virus/Pathogen DSP kit on a QIASymphony instrument as per the manufacturer’s instructions. WGS was performed using the NextSeq 550 platform and a NextSeq MID-output (2 × 150 bp)—paired-end sequencing kit. Libraries were prepared by following the Nextera XT Library preparation with Nextera XT indices. Reads were trimmed using the software (Trimmomatic v0.38; http://www.usadellab.org/cms/index.php) to remove sequencing adapters and low-quality bases [26]. FASTQC v0.11.4 was used to check the quality of raw and cleaned reads [27]. The de novo genome assembly of the isolates was performed on cleaned reads using SPAdes v3.12.0 [28]. The assemblies were checked with Quast v4.5 for the number of contigs and contig N50 [29]. All 37 isolates were retained as they passed the quality filters. The ARGs were predicted by the Antibiotic Resistance Genes Database (ARDB), the Comprehensive Antibiotic Resistance Database (CARD; https://card.mcmaster.ca; accessed on 23 May 2022) [30], and further confirmed by PointFinder database [31] and the ResFinder 4.1 EFSA 2021 database (accessed on 24 May 2022) [32]. We used the default thresholds for the detection of antimicrobial resistance genes. The percent identity and coverage for ResFinder and PointFinder were 95% and 60%, respectively. The description of the result was based on the PointFinder and ResFinder output. All isolate WGS reads are available in the SRA under BioProject PRJNA844571.

### 2.6. Statistical Analysis

A categorical table was generated with either a positive or a negative result for each isolate for each sampling point. Isolate susceptibility was first dichotomized as resistant or susceptible. The correlation between phenotypic and genotypic AMR was estimated using Pearson’s correlation coefficient. The analysis was performed using STATA version 15.0 (Stata Corporation, College Station, TX, USA) or the R Statistical Package version 4.0.0. The correlation was considered to be very high if *r* ≥ 0.90, high if *r* = 0.7 to 0.89, moderate if *r* = 0.5 to 0.69, low if *r* = 0.3 to 0.49, and negligible if *r* < 0.3 [33]. An isolate could have either concordant (phenotypic and genotypic resistance results agreed) or discordant (phenotypic and genotypic resistance results did not agree) outcomes. The statistical significance was set at *p* < 0.05. AMR profiles were described for each sample type.

## 3. Results

### 3.1. Prevalence of Antimicrobial Resistance at Entry into the Feedlot

GE isolates were recovered from 135/150 (90.0%) of feedlot entry samples. The MICs and resistance prevalence for each antimicrobial are presented in Table 2. None of the isolates were resistant to azithromycin, chloramphenicol, ciprofloxacin, gentamicin, nalidixic acid, or sulfisoxazole. The resistance prevalence to any one antimicrobial did not exceed 1/135 (0.7%) (i.e., very low). Only a single EE isolate was recovered from scant growth on an ESBL plate; this isolate exhibited resistance to amoxicillin/clavulanic acid, ampicillin, cefoxitin, ceftiofur, ceftriaxone, and tetracycline.

### 3.2. Prevalence of Antimicrobial Resistance at Slaughter

GE isolates were obtained from 129/150 (86.0%) post-slaughter fecal samples collected at the abattoir, and AMR was detected in 26/129 (20.1%) isolates. No resistance was observed for ciprofloxacin, gentamicin, and nalidixic acid. The most prevalent resistance was to tetracycline 23/129 (17.8%; moderate), followed by ampicillin 7/129 (5.4%; low), streptomycin 6/129 (4.7%; low), and sulfisoxazole 5/129 (3.9%; low). EE isolates were recovered from 6/150 (4.0%; low) samples, each obtained from scant growth (1–6 colonies) on ESBL agar plates. These isolates were all resistant to ampicillin, ceftiofur, and ceftriaxone (Table 3), and most were resistant to tetracycline 5/6 (83.3%), sulfisoxazole, and streptomycin 4/6 (66.7% each). No *E. coli* isolates were obtained from the hip (n = 150) or the flank and brisket (n = 149) carcass swab samples.

### 3.3. Resistance Profile Comparisons between Feedlot Entry and Slaughter

Most GE isolates obtained at the entry to the feedlot 133/135 (98.5%) and post-slaughter at the abattoir 103/129 (79.8%) were susceptible to all tested antimicrobials. At the entry to the feedlot, only a single GE isolate (0.7%; very low) was MDR (to ampicillin, streptomycin, and trimethoprim/sulfamethoxazole). For GE isolates recovered from abattoir fecal samples, six (4.6%) were MDR. ESBL-producing *E. coli* (at extremely low abundance) were recovered from one sample (0.7%) obtained at entry and six samples (4.0%) post-slaughter (Figure 1). All ESBL-producing *E. coli* isolates were resistant to ampicillin, ceftiofur, and ceftriaxone (Appendix A).

Of the study animals, most of them were Shorthorns (n = 81), and the rest were Angus (n = 30), Hereford (n = 22), Cross breed (n = 15) and Santa Gertrudis (n = 2). Of the sick cattle during the feeding period (n = 13), 12 of them were Shorthorns, and the remaining one was Angus (Appendix A). The *E. coli* isolated from the Shorthorn cattle breed was resistant to tetracycline (20%), sulfisoxazole (7.1%), streptomycin (5.7%), and ampicillin (4.3%). Isolates from Angus showed resistance to tetracycline (15.4%), ampicillin (11.5%) and streptomycin (7.7%). In *E. coli* isolated from Hereford, resistance to tetracycline (11.8%) and ampicillin (5.9%) was detected. Only tetracycline resistance (21.4%) was detected in *E. coli* isolated from cross breeds. On entry, the only ESBL-producing *E. coli* was detected from Angus. On exit, four of the ESBL-producing *E. coli* were isolated from Shorthorns, and the other two were from Hereford and cross breeds.

### 3.4. Detection of Antimicrobial Resistance Genes by Whole Genome Sequence Analysis

All phenotypically susceptible and 1/33 (3.0%) resistant *E. coli* isolates did not contain known ARGs. Of the resistant isolates, only one isolate returned a discordant outcome. Overall, 17 isolates possessed more than one ARG. Across all isolates, a total of 24 ARGs were observed, which conferred resistance to a range of antimicrobial classes including aminoglycosides, β-lactams, macrolides, folate synthesis inhibitors, phenicols, fluoroquinolones, and tetracyclines (Table 4). The most common ARGs observed in these isolates were *tet(B)* (n = 16, 43.2%), *aph(3″)-Ib* and *aph(6)-Id* (n = 12, 32.4%), *bla_TEM-1B_* (n = 9, 24.3%), and *sul2* (n = 9, 24.3%), which are responsible for resistance to tetracyclines, aminoglycosides, β-lactams, and sulfonamides, respectively. The most commonly detected β-lactamase ARGs were *bla_TEM_* (n = 11, 29.7%), *bla_CTX m_
*(n = 5, 13.5%), and *bla_CMY_* (n = 2, 5.4%). Two isolates harboured both *bla_CTX m_* and *bla_TEM_* genes. The combination of *bla_CMY_* and *bla_TEM_* was observed in only one isolate. For the EE isolates (n = 7), *bla_CTX m_* genes were detected in 5/7 (71.4%), *bla_TEM_* in 3/7 (42.8%) and *bla_CMY_* in 2/7 (28.6%) of the isolates. ARGs encoding reduced the susceptibility to ciprofloxacin (qnrS1), and resistance to chloramphenicol (cmlA1) and gentamicin (*aac(3)-IV*) was detected in a single isolate. However, the whole genome sequence analysis by PointFinder did not identify a mutation in the isolates.

The sequenced isolates harbored ARGs responsible for resistance to one (n = 15, 40.5%), two (n = 3, 8.1%), three (n = 10, 27.0%), four (n = 1, 2.7%), five (n = 2, 5.4%) or six (n = 1, 2.7%) antimicrobial classes (Table 5). Overall, 12/37 (32.4%) of the isolates contained ARGs responsible for classifying this cohort of isolates as MDR (resistance to one or more antimicrobials in three or more classes).

All β-lactam-resistant isolates harboured either *bla_TEM_*, *bla_CTX-M_*, or *bla_CMY_* ARGs. The same was true for azithromycin and chloramphenicol resistance phenotypes. Among the tetracycline-resistant isolates, 28/29 (96.5%) harbored either *tet(A)* or *tet(B)* ARGs. However, some ARGs were identified in isolates that were phenotypically susceptible. These included *aac(3)-IV*, *aph(3**″**)-Ib* and *aph(6)-Id*, *sul2*, and *dfrA5*, responsible for resistance to gentamycin, streptomycin, sulfisoxazole, and trimethoprim/sulfamethoxazole, respectively (one isolate each). By contrast, only a single tetracycline-resistant isolate harboured no associated resistance genes (Appendix A). Overall, resistance phenotypes were concordant with resistance genotypes (*r* ≥ 0.85: *p* < 0.05) (Figure 2, Appendix A).

## 4. Discussion

Surveys of AMR in commensal *E. coli* isolated from healthy animals is a common activity in food safety surveillance programs given that these isolates may act as reservoirs of ARGs with the potential to spread horizontally to other bacteria or serve as a direct source of infection [5]. The current study assessed the prevalence of AMR in *E. coli* isolates recovered from fecal samples obtained at induction and then post-slaughter at the abattoir for a cohort of 150 cattle from a single feedlot in South Australia. Two sources of *E. coli* were assayed; isolates derived from the most commonly occurring colony morphology on MAC (GE) and any isolates recovered on chromogenic ESBL agar (EE). This study had four major findings. First, the overall rates of resistance in GE isolates at feedlot entry were very low and only a single sample yielded an EE isolate. Second, although both overall resistance (1.5% to 20.1%) and multidrug resistance (0.7% to 4.6%) prevalence significantly increased in the post-slaughter fecal-derived GE isolates, the small increase in the recovery of EE isolates (0.7% to 4%) post-slaughter was not significant. Third, observed phenotypic resistance was highly correlated with the detected ARGs (n = 24) in the collection, including multiple ESBL genes in EE isolates. Fourth, no *E. coli* were isolated from either the hip (n = 150) or the flank and brisket (n = 149) swab samples, indicating effective hide removal, evisceration, and carcass hygiene and limited opportunity for fecal contamination of meat at the abattoir.

At feedlot entry, the majority of GE isolates showed susceptibility to the 14 tested antimicrobials, with only one MDR isolate recovered. These resistance frequencies were much lower than those reported in similar studies in Canada, where rates of resistance at entry into the feedlot were high for trimethoprim/sulfamethoxazole (44.4%) and ampicillin (20.3%), moderate for tetracycline (17.7–19.5%), and low for streptomycin (6.5%) [15,34]. To the best of our knowledge, there have been no other Australian studies that have surveyed AMR in commensal *E. coli* at feedlot entry. The low level of AMR at feedlot entry in our study may be influenced by strict guidelines for the use of antimicrobials in food-producing animals in Australia but more likely reflect low rates of antimicrobial use in the extensive grazing industry which represents over two-thirds of Australia’s beef production [16].

The most prevalent AMR detected in the post-slaughter GE isolates was tetracycline (17.8%), followed by ampicillin (5.4%), streptomycin (4.7%), and sulfisoxasole (3.9%). This moderate-to-low prevalence of resistance was to antimicrobials considered to be of low importance in Australian health care settings (ASTAG, 2018). The resistance rates were similar to a large post-slaughter cross-sectional Australian beef cattle study, where feedlot cattle isolates were observed to have a similar level of tetracycline resistance (15%) compared to grass-fed cattle (2.6 to 4.6%), with no resistance to fluoroquinolones or third-generation cephalosporins identified [35].

The 24 ARGs identified in the *E. coli* collection by WGS conferred resistance to a range of antimicrobial classes including aminoglycosides, β-lactams, fluoroquinolones, folate synthesis inhibitors, macrolides, phenicols, and tetracyclines. Resistance was most commonly detected to antimicrobial classes that have been historically used but are no longer registered in food-producing animals (e.g., streptomycin and chloramphenicol) or have experienced widespread and continuous use as animal treatments (e.g., tetracyclines and trimethoprim/sulfonamides) [36]. Despite fluoroquinolones not being registered for use in Australian food-producing animals, a plasmid-mediated quinolone ARG (*qnrS1*) was found in a single MDR EE isolate, which has been reported in other Australian livestock-focused AMR survey studies [9,21].

Resistance to extended-spectrum β-lactam antimicrobials is a major concern worldwide, as these antimicrobials are critically important for both human and veterinary medicine [37]. Resistance to extended-spectrum β-lactams is usually due to the production of extended-spectrum and AmpC β-lactamases, which can inactivate many newer generation β-lactam antimicrobials [38,39]. The use of third- and fourth-generation cephalosporins is often linked with the detection of ESBL/AmpC-producing bacteria in healthy food-producing animals [40]. In this study, the recovery of ESBL-producing *E. coli* (albeit at extremely low abundances on the ESBL plate) increased from one sample (0.7%) at the entry to six samples (4.0%) at the abattoir. This result showed that cattle entering feedlots may already be colonised with commensal *E. coli* resistant to critically important antimicrobials (i.e., ceftiofur), albeit at extremely low frequency and abundance. The differences in prevalence in both sampling points may have occurred by chance or possibly associated with the dissemination of the resistant isolate from the carrier cattle, or the exposure of the cattle to individuals admitted to the hospital pen for antimicrobial treatment, where ceftiofur is used as a treatment for cases that do not respond to lower importance drugs, or where there is a risk of exceeding the export slaughter interval [20]. By comparison, a significantly higher frequency of resistance to third-generation cephalosporins (35%) was reported in *E. coli* isolated from beef cattle in Germany [41]. The observed difference in the prevalence of AMR bacteria could be due to the variation in beef production systems and antimicrobial use.

One or more resistance genes responsible for the observed resistance phenotypes were detected by whole genome sequence analysis. A range of ESBL resistance genes were detected, including *bla_TEM-1B_* 8/37 (21.6%), *bla_CTX-M-15_*, and *bla_CTX-M-27_ 3/37* (8.1%, each), *bla_CMY-2_*, and *bla_TEM-1C_* 2/37 (5.4%, each). From EE isolates, the *bla_CTX-M-15_* and *bla_TEM-1B_* resistance genes were detected in 3/7 (42.8%, each) followed by *bla_CMY-2_* and *bla_CTX-M-27_* in 2/7 (28.6%, each). These ESBL and AmpC beta-lactamase genes have been previously described in livestock isolates from international studies as well as from several conducted in Australia [9,42,43,44]. These genes are usually found within *E. coli* clonal lineages that have a broad host range, such as ST10, ST58, ST131, ST155, or ST3891, having been isolated from humans, animals, and the environment throughout the world [9,43,45]. Due to the potential transfer of such strains to humans via the food chain and environment, the occurrence of ESBL-producing *E. coli* in food-producing animals is a public health concern; however, the results of the present study confirm that in Australia, they are likely to be found in the gut of healthy Australian cattle pre- and post-feedlot entry at extremely low (<10 CFU/g of fecal matter) populations levels. Sequence types and clonal relationships will now be explored through further interrogation of the genome in a follow-up study.

Oxytetracycline and chlortetracycline are broad-spectrum antimicrobials commonly used to treat bacterial infection in feedlot cattle in Australia [20]. Tetracycline resistance mainly occurs through efflux pumps, enzymatic inactivation, or ribosomal protection proteins [46] and ARGs involved in these resistance mechanisms are often encoded on mobile genetic elements. In the present study, both *tetA* and *tetB* genes were detected in 36.4% and 48.5% of the sequenced antimicrobial-resistant isolates, respectively, as well as 96.5% of the tetracycline-resistant isolates. The relatively moderate increase in phenotypic tetracycline resistance from feedlot entry (0%) to exit (17.8%) in this longitudinal study suggests a possible role for antimicrobial selection pressure as well as additional undefined mechanisms from the feedlot environment are driving carriages (e.g., previous metaphylactic treatment, co-selection of resistance, pen to pen transfer of resistance or frequent mobile genetic element transmission and carriage), as only 1 animal out of 150 was treated therapeutically with oxytetracycline during the feeding period.

Resistance to aminoglycoside antimicrobials mainly occurs due to the production of aminoglycoside-modifying enzymes (AME) such as adenylyl transferases (ANTs), O-phosphoryl transferases (APHs), and *N*-acetyl transferases (AACs) [47]. In this study, multiple aminoglycoside-modifying, enzyme-encoding genes such as *aph(3″)-Ib* and *aph(6)-Id* (32.4% each), and *ant(3″)-Ia*, *aph(3′)-Ia*, *aph(4)-Ia*, and *aac(3)-IV* (2.7% each) were detected in phenotypically resistant isolates. It is reported that plasmid-mediated genes which confer aminoglycoside resistance are widely distributed in the environment [48], given the widespread historic use of streptomycin.

In this study, the correlation between phenotypes and genotype AMR was explored by an MIC susceptibility test and WGS. The AMR phenotype and possession of corresponding ARGs were in agreement in the majority of cases. Overall, AMR at entry into the feedlot was lower compared to that of abattoir isolates. However, the main limitation of this study was that it was conducted over a single feedlot animal rotation from one beef feedlot farm in Australia. In addition, our intention to assess the effect of treatment on the development of AMR was not successful. The number of treated cattle individuals (n = 13) was small for any meaningful analysis at the level of the administered antimicrobial. A prospective cohort study would be required to assess directly the effect of the administration of a particular antimicrobial on the resistance profile of fecal *E. coli*. Further large-scale longitudinal studies in beef cattle production systems will be required to more comprehensively detail changes in bacterial AMR status. The link between AMR, plasmids, virulence factors, and other mobile genetic elements should be determined. Further research is also required to determine if the antimicrobial resistance profile and phylogenetic origins of commensal *E. coli* inhabiting the ruminant alimentary tract are influenced by anthropozoonotic, environmental, or local feedlot factors including exposure to antimicrobials.

## 5. Conclusions

This study determined the AMR profile of *E. coli* from beef cattle over time from the entry into the feedlot to the abattoir. The cattle arrived at the feedlot with relatively low numbers of AMR bacteria compared to the exit. Overall, in this study, a low level of AMR was detected, and it was hypothesised that the development and spread of AMR in *E. coli* in beef feedlot cattle might be influenced by factors other than antimicrobial treatment including feed, environment, farm type, management, and other factors. Extensive and coordinated surveillance is a critical requirement for the efforts to control antimicrobial resistance. Routine monitoring would allow for the timely detection of emerging and existing modes of resistance and genes of AMR in bacteria from food-producing animals, including beef feedlot systems.

## Figures and Tables

**Figure 1 animals-12-02256-f001:**
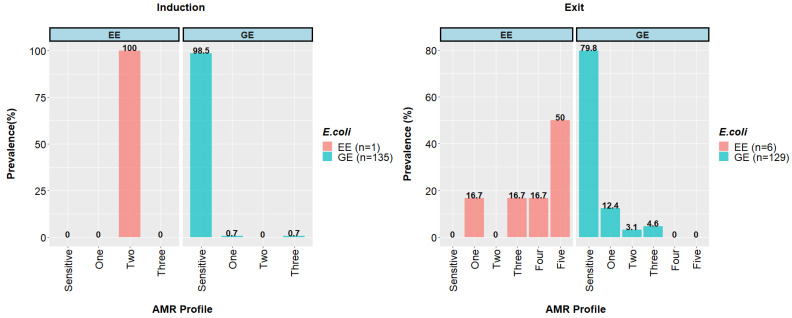
Antimicrobial resistance profiles of Escherichia coli recovered from MacConkey agar (GE) and ESBL agar (EE) at entry into the feedlot (Induction) and at the abattoir post-slaughter (Exit). GE, generic *E. coli*; EE, ESBL-producing *E. coli*. Only two (1.5%; low) feedlot entry GE isolates were resistant to at least one antimicrobial. The single (0.7%; very low) EE isolate obtained at feedlot entry exhibited resistance to β-lactams (amoxicillin/clavulanic acid, ampicillin, cefoxitin, ceftiofur, and ceftriaxone) and tetracycline. By comparison, the six EE isolates obtained post-slaughter were resistant to one (16.7%), three (16.7%), four (16.7%), and five (50%) antimicrobial classes.

**Figure 2 animals-12-02256-f002:**
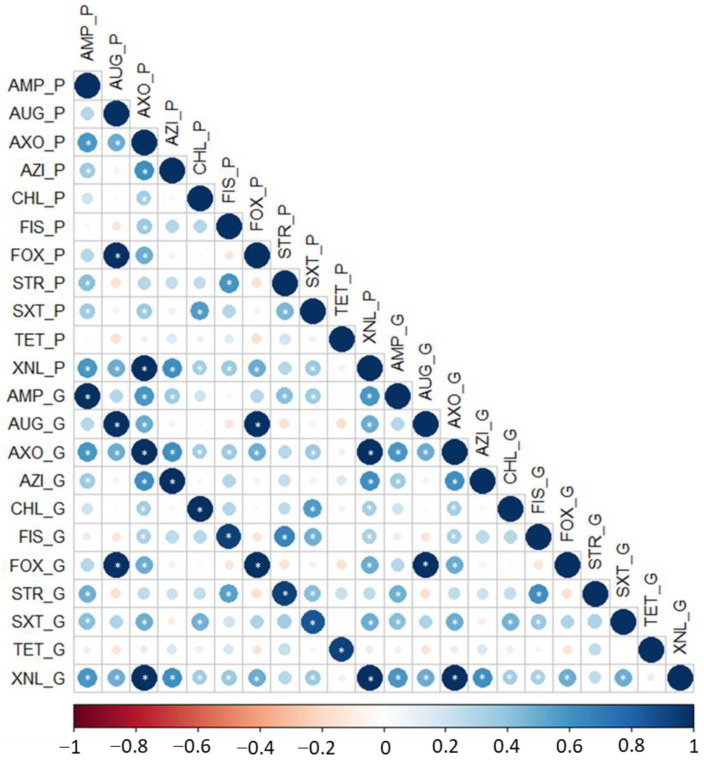
The Pearson correlation coefficient matrix between phenotypic (P) and genotypic (G) AMR in Escherichia coli isolated from beef feedlot and abattoir. AUG, amoxicillin/clavulanic acid; AMP, ampicillin; AZI, azithromycin; AXO, ceftriaxone; CHL, chloramphenicol; FIS, sulfisoxazole; FOX, cefoxitin; STR, streptomycin; TET, tetracycline; SXT, trimethoprim/sulfamethoxazole; XNL, ceftiofur. According to the scale bar shown at the bottom, the colours indicate different correlation coefficient values. The circle size is proportional to the correlation coefficients. Asterisks indicate the significance of the Pearson correlation coefficient (*p* < 0.05).

**Table 1 animals-12-02256-t001:** Tested dilution ranges and breakpoints used for antimicrobial susceptibility testing.

Antimicrobial Agent	Tested Range	Breakpoints
Amoxicillin/Clavulanic acid	1–32	≥32
Ampicillin	1–32	≥32
Azithromycin	0.12–16	>16
Cefoxitin	0.5–32	≥32
Ceftiofur	0.12–8	≥8
Ceftriaxone	0.25–64	≥4
Chloramphenicol	2–32	≥32
Ciprofloxacin	0.015–4	≥1
Gentamicin	0.25–16	≥16
Nalidixic acid	0.5–32	≥32
Streptomycin	2–64	≥64
Sulfisoxazole	16–256	> 256
Tetracycline	4–32	≥16
Trimethoprim/Sulfamethoxazole	0.12–4	≥4

**Table 2 animals-12-02256-t002:** Antimicrobial susceptibility testing results for generic Escherichia coli recovered on MacConkey agar from cattle rectal feces obtained at entry into the feedlot (n = 135).

Antimicrobial Class	Antimicrobial Agent	Prevalence (95% CI)	Isolate Prevalence (%)^*^ for Each MIC Value Tested (µg/mL)															
			0.015	0.03	0.06	0.12	0.25	0.5	1	2	4	8	16	32	64	128	256	512
Aminoglycosides	Gentamicin	0.0 (0.00–2.77)						63.7	35.6	0.7								
	Streptomycin	0.7 (0.13–4.08)									55.6	43.0	0.7		0.7			
β-lactams	Ampicillin	0.7 (0.13–4.08)							6.7	46.7	43.7	0.7	1.5	0.7				
	Amoxicillin/Clavulanic acid	0.7 (0.13–4.08)							3.7	15.6	65.9	13.3	0.7	0.7				
	Cefoxitin	0.0 (0.00–2.77)								26.7	64.4	8.9						
	Ceftiofur	0.0 (0.00–2.77)				0.7	14.1	79.2	5.2	0.7								
	Ceftriaxone	0.0 (0.00–2.77)					97	1.5	1.5									
Folate pathway inhibitor/antagonists	Sulfisoxazole	0.0 (0.00–2.77)											88.1	11.0			0.7	
	Trimethoprim/ Sulfamethoxazole	0.7 (0.13–4.08)				98.5	0.7				0.7							
Macrolides	Azithromycin	0.0 (0.00–2.77)						3.0	15.6	36.3	44.4		0.7					
Phenicols	Chloramphenicol	0.0 (0.00–2.77)								3.0	48.9	47.4	0.7					
Fluoroquinolones	Ciprofloxacin	0.0 (0.00–2.77)	97.8	1.5			0.7											
	Nalidixic acid	0.0 (0.00–2.77)						0.7	13.3	81.5	4.4							
Tetracycline	Tetracycline	0.0 (0.00–2.77)									99.3							

* The white area shows the dilution range, and the shaded areas are MICs beyond the concentration tested for each antimicrobial. Solid vertical lines show the breakpoints used for classifying an isolate as resistant.

**Table 3 animals-12-02256-t003:** Antimicrobial susceptibility testing results for generic Escherichia coli (n = 129; GE) and ESBL-producing *E. coli* (n = 6; EE) isolated from rectal fecal samples collected at the abattoir post-slaughter.

Antimicrobial Class	Antimicrobial Agent	*E. coli* Isolate Type	Prevalence (95% CI)	Isolate Prevalence (%) * for Each MIC Value Tested (µg/mL)															
				0.015	0.03	0.06	0.12	0.25	0.5	1	2	4	8	16	32	64	128	256	512
Aminoglycosides	Gentamicin	GE	0.0 (0.00–2.89)					3.9	72.8	23.2									
		EE	0.0 (0.00–39.03)						66.7	16.7			16.7						
	Streptomycin	GE	4.7 (1.16–16.8)									53.5	39.5	1.5	0.8	3.9	0.8		
		EE	66.7 (53.51–75.88)									33.3				33.3	33.3		
β–lactams	Ampicillin	GE	5.4 (1.34–20.02)							15.5	47.3	28.7	2.3	0.8		5.4			
		EE	100.0 (96.00–100.00)													100			
	Amoxicillin/Clavulanic acid	GE	0.8 (0.11–5.75)							6.2	35.6	47.3	10.1		0.8				
		EE	16.7 (8.16–31.05)									33.3	50.0		16.7				
	Cefoxitin	GE	0.8 (0.11–5.75)							11.6	31.8	36.4	19.4			0.8			
		EE	16.7 (8.16–31.05)									66.7	16.7			16.7			
	Ceftiofur	GE	0.0 (0.00–2.89)				12.4	28.7	52.7	6.2									
		EE	100.0 (96.0–100.0)											100.0					
	Ceftriaxone	GE	0.0 (0.00–2.89)					100.0											
		EE	100.0 (96.0–100.0)												16.7		83.3		
Folate pathway inhibitor/antagonists	Sulfisoxazole	GE	3.9 (1.82–8.45)											92.2	3.9				3.9
		EE	66.7 (53.51–75.88)											33.3					66.7
	Trimethoprim/ Sulfamethoxazole	GE	0.0 (0.00–2.89)				100												
		EE	33.3 (23.68–45.70)				33.3	33.3					33.3						
Macrolides	Azithromycin	GE	0.0 (0.00–2.89)							10.1	38.7	51.2							
		EE	50.0 (34.83–65.64)								33.3	16.7			50.0				
Phenicols	Chloramphenicol	GE	0.0 (0.00–2.89)								0.8	32.6	65.1	1.5					
		EE	16.7 (8.16–31.05)										66.7	16.7	16.7				
Fluoroquinolones	Ciprofloxacin	GE	0.0 (0.00–2.89)	96.9	3.1														
		EE	0.0 (0.00–39.03	66.7	16.7				16.7										
	Nalidixic acid	GE	0.0 (0.00–2.89)							11.6	77.5	10.8							
		EE	0.0 (0.00–39.03)								50.0	33.3	16.7						
Tetracycline	Tetracycline	GE	17.8 (12.66–24.43)									81.4	0.8		2.3	15.5			
		EE	83.3 (67.50–93.75)									16.7				83.3			

* The white area shows the dilution range, and the shaded areas are MICs beyond the concentration tested for each antimicrobial. Solid vertical lines show the breakpoints used for classifying an isolate as resistant.

**Table 4 animals-12-02256-t004:** The identification of antimicrobial resistance genes in Escherichia coli isolates (n = 37) recovered from fecal samples obtained at entry to the feedlot and post-slaughter at the abattoir.

Antimicrobial Class	Resistance Phenotype	Resistance Gene	Number of Isolates (%)
Aminoglycosides	STR	*aph(3″)-Ib*	12 (32.4)
Aminoglycosides	STR	*aph(6)-Id*	12 (32.4)
Aminoglycosides	STR, KAN	*ant(3″)-Ia*	1 (2.7)
Aminoglycosides	KAN	*aph(3′)-Ia*	1 (2.7)
Aminoglycosides	HYG	*aph(4)-Ia*	1 (2.7)
Aminoglycosides	GEN	*aac(3)-IV*	1 (2.7)
β-lactams	AUG, AXO, FOX, XNL	*bla_CMY-2_*	2 (5.4)
β-lactams	AMP, AXO, XNL	*bla_CTX-M-15_*	3 (8.1)
β-lactams	AMP, AXO, XNL	*bla_CTX-M-27_*	2 (5.4)
β-lactams	AMP, AXO, XNL	*bla_TEM-1B_*	9 (24.3)
β-lactams	AMP	*bla_TEM-1C_*	2 (5.4)
Macrolides	AZI, ERY	*mph(A)*	3 (8.1)
Macrolides	AZI, ERY	*mph(E)*	1 (2.7)
Macrolides	AZI, ERY	*msr(E)*	1 (2.7)
Folate synthesis inhibitors	FIS	*sul1*	1 (2.7)
Folate synthesis inhibitors	FIS	*sul2*	9 (24.3)
Folate synthesis inhibitors	FIS	*sul3*	1 (2.7)
Folate synthesis inhibitors	SXT	*dfrA5*	2 (5.4)
Folate synthesis inhibitors	SXT	*dfrA12*	1 (2.7)
Folate synthesis inhibitors	SXT	*dfrA14*	1 (2.7)
Phenicols	CHL	*cmlA1*	1 (2.7)
Fluoroquinolones	CIP	*qnrS1*	1 (2.7)
Tetracyclines	TET	*tet(A)*	12 (32.4)
Tetracyclines	TET	*tet(B)*	16 (43.2)

AUG, amoxicillin/clavulanic acid; AMP, ampicillin; AZI, azithromycin; AXO, ceftriaxone; CHL, chloramphenicol; ERY, erythromycin; FIS, sulfisoxazole; FOX, cefoxitin; GEN; gentamicin; HYG, hygromycin; STR, streptomycin; TET, tetracycline; SXT, trimethoprim/sulfamethoxazole; XNL, ceftiofur.

**Table 5 animals-12-02256-t005:** The antimicrobial resistance pattern of *E. coli* recovered from fecal samples obtained at entry to the feedlot and post-slaughter.

Antimicrobial Classes Pattern	Total no. of Isolates (%)		Resistance Pattern (no. of Isolates)	
	Phenotypic (37)	Genotypic (37)	Phenotypic (MIC)	Genotypic (Resistance Gene)
All susceptible	4 (10.8)	5 (13.5)	4	5
1	16 (43.2)	15 (40.5)	AMP (1)	*bla_TEM-1B_* (1)
			AMP-AUG-FOX-XNL-AXO (1)	*bla_CMY-2_* (1)
			FIS (1)	*sul2* (1)
			TET (13)	*tet(A)*(4)
				*tet(B)* (8)
2	5 (13.5)	3 (8.1)	AMP-AUG-AXO-FOX-TET-XNL (1)	*bla_TEM-1C_*, *tet(A)* (2)
			AMP-TET (3)	*sul2*, *tet(B)* (1)
			FIS-TET (1)	
3	8 (21.6)	10 (27.0)	AMP-STR-SXT (1)	*bla_CMY-2_*, *bla_TEM-1B_*, *dfrA5*, *tet(A)* (1)
			AMP-STR-TET (3)	*bla_CTX-M-15_*, *mph(E)*, *mph(A)*, *msr(E)*, *tet(A)* (1)
			AMP-AXO-AZI-TET-XNL (1)	*aph(3″)-Ib*, *aph(6)-Id*, *bla_TEM-1B_*, *tet(B)* (4)
			FIS-STR-TET (3)	*aph(3′)-Ia*, *aph(3″)-Ib*, *aph(6)-Id*, *bla_TEM-1B_*, *dfrA5*, *sul2* (1)
				*aph(3″)-Ib*, *aph(6)-Id*, *sul2*, *tet(B)* (3)
4	1 (2.7)	1 (2.7)	AMP-AXO-FIS-STR-SXT-TET-XNL (1)	aph(3″)-Ib, aph(6)-Id, bla_CTX-M-15_, bla_TEM-1B_,dfrA14, sul2, tet(A) (1)
5	3 (8.1)	2(5.4)	AMP-AXO-AZI-FIS-STR-TET-XNL (2)	aph(3″)-Ib, aph(6)-Id, bla_CTX-M-27_,mph(A), sul2, tet(A) (2)
			AMP-AXO-CHL-FIS-STR-SXT-TET-XNL (1)	
6		1 (2.7)		aac(3)-IV, ant(3″)-Ia, aph(3″)-Ib, aph(4)-Ia, aph(6)-Id, bla_CTX-M-15_, bla_TEM-1B_, cmlA1, dfrA12, qnrS1, sul1, sul3,tet(A) (1)
Non-MDR	21 (56.7)	18 (48.6)		
MDR	12 (32.4)	14 (37.8)		
Resistance	33 (89.2)	32(86.5)		

AUG, amoxicillin/clavulanic acid; AMP, ampicillin; AZI, azithromycin; AXO, ceftriaxone; CHL, chloramphenicol; FIS, sulfisoxazole; FOX, cefoxitin; STR, streptomycin; TET, tetracycline; SXT, trimethoprim/sulfamethoxazole; XNL, ceftiofur.

## Data Availability

All isolate WGS reads are available in the SRA under BioProject PRJNA844571.

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
