# Peer review of "Phenotypic and Genotypic Analysis of Antimicrobial Resistance in Escherichia coli Recovered from Feedlot Beef Cattle in Australia"

_animals, 2022, doi:10.3390/ani12172256_

Round 1
Reviewer 1 Report
This study looks into the phenotypic and genotypic analysis of AMR in E. coli in feedlot cattle and determined the overall level of antimicrobial resistance burden. the corresponding ARGs were also determined.
However, there are some comments that need to be clarified (as attached). Overall this paper is well written.

Author Response
Thank you for your constructive comments

Reviewer 2 Report
This manuscript provides evidence of AMR in the feedlot industry in Australia and it is a relevant topic to be disseminated. The manuscript is overall well-written and very well presented. Only a few notes for authors that may be helpful for improvement, although the overall quality is very good, congratulations!
- Did you compare the different AMR within the breeds?
- What is the specific purpose of selecting 13 cattle for antimicrobial treatment? Did you think inclusion of these 13 animals just disturb the prevalence data as they received antimicrobial treatment that would likely contribute to higher resistance risk?
- Did the animals receive any regular medicinal treatment/ antimicrobials during in the feedlot?
- How the 150 animals were determined? Any criteria in the enrolment?
- L358: but it was not observed at the entry point. But I agree that the observed AMR might be from individuals that admitted to the hospital pen. Then, it would be good to separate or exclude the 13 animals from the analysis?
Author Response

(The authors gave the same response as above.)

Reviewer 3 Report
The Study, focuses on antimicrobial resistance in cattle, using E.coli – an indicator organism. The phenotypic resistance (MIC) and detection of antimicrobial resistance genes (WGS-based) were assessed at two-time points - entering and exiting the feedlot. The title is consistent with the subject of the manuscript. The appropriate methodology was applied. The whole manuscript has fitting composition. Conclusions are supported by the results, where the Authors confirmed the spread of antimicrobial resistance at the exit point of the study.
Nevertheless, the Author should consider several problematic areas of the Manuscript;
· While "Introduction" provides significant background for the whole study, the Authors may consider shortening it, thus making it more synthetic and to the point.
· Authors should add the date of the update for each of the databases and thresholds (for both coverage and identity) with which the analysis was performed. The Applied database should be as current as possible.
· Authors several times undertook the subject of mobilization of resistance genes. However, detection of plasmid markers was not performed. Maybe They should consider adding this data to the manuscript. Confirming the presence of plasmid replicons (e.g. using the PlasmidFinder tool) on the same contig as the AMR gene may further support the thesis about resistance mobilization via plasmids.
· In methodology CARD and ResFinder are mentioned as applied tools/databases. The result of AMR detection (nomenclature of beta-lactamases gene in particular) between tools may vary. There is no mention of whether the obtained results were the same or different using both approaches. However, if the description of the results was based only on one of the above-mentioned tools, the methodology should reflect that.
· According to the methodology data set was checked for point mutation. However, no information considering this was presented in the "Results". Even if point mutation were not present, this should be mentioned, because it is valuable information.
· Authors very well describe MIC results in text and using Figure 1. Thus, in the reviewer’s opinion Tables, no. 2 and 3 should be moved to "Supplementary", to avoid duplication of presented data.
· The reviewer suggests rearranging the composition of Table no. 4 - starting with resistance phenotype rather than genotype. Also, the Authors should consider combining Tables no. 4 and 5 to avoid repetitions of data.
· Authors incorrectly classified detected blaTEM variants -1B and -1C as ESBL or/and AmpC beta-lactamase (line 370-372). Some TEM variants (usually with higher order numbers) may code resistance to cephalosporins (ESBL or AmpC phenotype). However, genes detected by the Authors are responsible only for resistance to beta-lactams
· Table S2 contains very similar information as Table no. 4 and no. 5. In the reviewer's opinion able S2 should be more detailed. Present configuration unable to check some of the data described in the Manuscript – e.g. cases of AMR co-resistance in the context of a specific isolate. Maybe Authors can consider arranging S2 according to the isolate
· Line 239 – incorrect font formatting
Author Response

(The authors gave the same response as above.)
